## [Transparent Peer Review file · Nature Communications]

ATRX loss couples genome instability at a G-rich repeat to dysregulation of human alpha-globin expression

Corresponding Author: Professor Douglas Higgs

Version 0:

Reviewer comments:

Reviewer #1

(Remarks to the Author)

This manuscript from Shen et al studies the role of ATRX in regulating transcriptional output from the α -globin locus. The main conclusion of this work is that ATRX enhances expression of the α -like globin gene HBM by resolving R-loop formation and DNA damage at a G-rich variable number tandem repeat (VNTR) DNA sequence near the gene promoter. Overall, the concepts presented are not novel per se, rather the strength of the study is that it confirms previous literature and presents a unified theory for ATRX function in regulating gene expression from the α -globin locus. The study itself is well done, using genetic and chemical perturbation to probe the role of the VNTR in HBM expression. However, the conceptual advances are modest, the scope in studying a single locus is limited, some claims are poorly supported, and some observations remain unexplained.

The work follows up on a seminal finding from the same lab that ATRX plays a role in regulating genes near tandem repeats, with longer VNTRs resulting in a higher degree of gene downregulation upon ATRX loss (PMID 21029860). In the current study, the authors use CRISPR gene editing tools to delete the VNTR proximal to HBM and demonstrate that HBM expression is elevated and no longer dependent upon ATRX. This part of the study extends upon a study drawing similar conclusions for the HBM promoter in the context of a luciferase reporter assay (PMID 26991472). The current study goes on to link ATRX to G-quadruplex binding, R-loop formation, DNA damage, and transcription. Many studies in the literature have demonstrated such connections (e.g., PMID 21029860, 25452430, 28487353, 30808951, 34162889, 35013239).

One conclusion from this work is that ATRX deficiency downregulates α -globin in a subset of cells exhibiting DNA damage. This part of the work is poorly supported, and the authors do not definitively demonstrate that the cells exhibiting signs of DNA damage are the same cells downregulating α -globin transcription. Further, the authors do not explain why loss of ATRX leads to stochastic changes in α -globin gene expression on a single cell level. These claims and observations require additional experimental support and analysis.

Specific Comments:

1. The authors state that only a subset of ATRX KO BFU-E clones show decreased HBM and HBA expression. First, the authors are missing callout to Extended Data Fig. 1j, which shows this relationship better than the box plots (where information about single cell behavior is lost) chosen for the main figure Fig. 1f. This point should be better represented in the main figure, to show the expression level of HBM and HBA in individual BFU-E, compared to their ATRX expression level.
2. Do all 53 downregulated genes in ATRX KO cells have VNTRs near their promoters or regulatory elements? Such genome-wide analysis would help to expand the scope of the study and the impact of the conclusions.
3. Lines 167-169 – links to DNA damage and HBM downregulation from the single cell data are weak. The argument is that (1) some ATRX KO cells have increased RNF168 expression; (2) some ATRX KO cells have increased gamma-H2AX staining; and (3) some ATRX KO cells with low HBM expression show an ontology term for histone H2A ubiquitination pathways. The authors have not confidently demonstrated that any of these observations are linked. Related, lines 313-315 claim in discussion that the work demonstrates increased H2A ubiquitylation in ATRX KO is overstated. The authors identify this pathway as an ontology term but they do not directly measure H2AUb.
4. The authors do not directly measure DNA damage at the α -globin VNTR in the absence of ATRX but rather make inferences that DNA damage of this locus plays a role in its transcriptional repression. The location of DNA damage in the

absence of ATRX should be assessed.

5. Out of curiosity, what happened to tubulin at the 4 hr time point in Extended Data Fig. 4c?

Reviewer #2

(Remarks to the Author)

In this manuscript Shen et al examine ATRX function in gene regulation in an erythroid differentiation model. ATRX mutations are known to result in a mild form of alpha thalassemia. The goal of this paper is to develop a mechanistic understanding of ATRX function in this process. First the authors show that loss of ATRX results in reduced HBM expression only in a subset of cells. They further perform single cell RNA seq to show deregulation in a subset of cells and that a subset of these cells also exhibit gamma H2AX foci indicative of DNA damage. In addition RNA sequencing analysis showed an upregulation of DNA damage response pathways. They develop a degron model where they can acutely deplete ATRX in HUDEP2 cells. Using these cells they show that ATRX binds to the G-rich VNTR region at the alpha globin locus in a transcription dependent manner and that loss of ATRX results in reduced HBM expression. Effects of ATRX loss are attenuated by deletion of the VNTR region. Finally they show that the VNTR region accumulates R-loops in the absence of ATRX. Interestingly introduction of DNA damage at these sites also recapitulates ATRX loss with respect to gene expression.

Overall this is a straightforward paper that is easy to follow and logical in narrative. However most of the conclusions of this paper have already been published previously by the same group or by several different groups in other journals as mentioned below. While the degron system itself is novel, there is very little novelty in the experiments performed using this system. Overall this manuscript does not contribute significantly to advancing our understanding of how ATRX regulates gene expression beyond what we know already.

Specific comments.

1. Several groups have already shown that ATRX loss results in accumulation of DNA damage in different cell types. These include Wong et al in genome research 2010, Lovejoy et al plos genetics 2012, Huh et al JCI 2012. This is not new or unexpected.
2. Gene ontology analysis in a small set of genes is not very informative or accurate. Presence of one or two gene in a category can point to the enrichment of a pathway. The same conclusions can be obtained by just looking at the gene names.
3. Fig 4 - Does DNA damage actually occur at the VNTR upon ATRX loss and R-loop accumulation? Some genomic method to evaluate this is needed.
4. Fig 4c,d - Does inducing DNA damage cause R-loop formation at VNTRs given the recent reports of R-loops in DNA damage repair?
5. The title of the paper is misleading - it suggests that ATRX is able to resolve R-loops, when there is no evidence to suggest that ATRX functions in unwinding R-loop or G4 structures.

Reviewer #3

(Remarks to the Author)

This study by Shen et al investigates the molecular mechanism by which mutations in the chromatin remodeler ATRX cause α -thalassemia, a feature of the developmental disorder ATR-X syndrome. The authors propose a multi-step mechanism for the regulation of the α -globin gene: ATRX deficiency leads to the accumulation of R-loops and G-quadruplex (G4) structures at a specific G-rich variable number tandem repeat (VNTR) in the α -globin locus, which in turn induces local DNA damage and subsequent transcriptional silencing of the adjacent α -globin genes.

They employ complementary experimental systems including CRISPR-edited primary CD34+ hematopoietic stem cells and an inducible ATRX degron system in HUDEP-2 cells. The former approach allows for physiologically relevant analysis in primary cells and the latter enabled temporal control of ATRX depletion. The scRNAseq analysis is particularly valuable, as it revealed the stochastic nature of the α -globin downregulation. The most compelling aspect of the study is the genetic rescue and recapitulation experiments.

Nevertheless, several weaknesses should be addressed:

1-Several figures lack statistical testing or only show representative data without quantification. The sample sizes for some experiments is low (some even at n=2) and appear limited for drawing any conclusion. The authors should include rigorous statistics to all quantitative assays: replicate numbers (replicates should be biological rather than technical), adjusted p-values, and clear indication of the statistical tests used.

2- While the paper establishes that ATRX loss leads to R-loop accumulation and subsequent DNA damage, it remains unclear whether ATRX primarily resolves R-loops, which then stabilize G4s, or acts on both independently. Additional experiments, such as overexpressing RNase H1 to degrade R-loops, could clarify whether resolving R-loops alone prevents DNA damage and rescues α -globin expression.

3- The authors suggest that the mechanism may be general, citing downregulation of NME4, another VNTR-containing gene. However, this is based on limited evidence. A genome-wide analysis of ATRX binding at G-rich repeats, R-loop

accumulation, and transcriptional effects upon ATRX loss are required to support claims of broader relevance.

4- The paper does not fully address why only a subset of ATRX-deficient cells exhibit the phenotype. Factors such as differences in transcription levels, replication timing, or pre-existing epigenetic states at the locus should be investigated to provide deeper insight into the phenotype's variability. The authors should investigate whether transcriptional load, replication timing or local chromatin marks or histone variants predict which ATRX-null cells become HBMlow.

5- While the paper cites studies supporting its model of R-loop and G4-mediated DNA damage, it does not adequately address alternative or conflicting mechanisms of ATRX-mediated α -globin regulation. For instance, the role of ATRX in histone variant exchange, particularly the negative regulation of macroH2A incorporation at the α -globin locus, is well-documented in the literature. This study shows that ATRX loss leads to macroH2A accumulation at the HBA cluster, contributing to transcriptional repression. Although the current paper briefly mentions histone H2A ubiquitination pathways and cites Ratnakumar et al., 2012 in the discussion, it does not fully integrate or discuss this alternative pathway as a potential complementary or competing mechanism. A more explicit acknowledgment or comparison would improve the manuscript.

6- The temporal correlation between HBZP1 transcription and the appearance of an ATRX ChIP-seq peak at the $\psi\zeta$ VNTR, on its own, is not sufficient to conclude that ATRX binding is transcription-dependent. Differentiation can potentially change many variables—chromatin accessibility, histone modifications or variants, replication timing, DNA topology and the abundance of ATRX-binding partners. Any one of these could, in principle, be the true driver of ATRX recruitment, with HBZP1 activation occurring in parallel.

7- Several findings consolidate or refine previously reported associations and consequently the study provides important, but at times incremental information (ATRX–VNTR binding, G4 propensity, macroH2A antagonism). The binding of ATRX at this VNTR was previously reported. The G4 aspect has been described. The use of CX-5461 confirms, rather than discovers, the G4 component. MacroH2A and other epigenetic players are mentioned but not mechanistically explored; Ratnakumar et al. already implicated macroH2A in α -globin repression after ATRX loss.

Reviewer #4

(Remarks to the Author)

In this manuscript, Shen et al. demonstrated that ATRX is an essential regulator for alpha-globin gene expression through resolving R-loop formation and preventing DNA damage at the G-rich VNTR located upstream of HBM gene. The pioneer work from Dr. Higg's lab revealed that in ATR-X syndrome, mutation of ATRX resulted in alpha-thalassemia through downregulated alpha-globin genes. In this study, they provide a unique mechanism by which ATRX specifically regulates alpha-globin gene expression via preventing persistent R-loop formation in upstream of HBM gene. The manuscript is well written, and the conclusions are generally supported by the experiments conducted by the authors. However, there are some comments that need to be addressed to strengthen and improve the manuscript.

1. Figure 1 showed that ATRX KO efficiency is more than 70%, yet HBM is reduced only in very small fraction of cells. Majority of ATRX KO cells do not exhibit reduction of alpha-globin genes. Are variable length and repeat # of VNTR involved in regulation of ATRX dependent alpha globin regulation? Does VNTR variable affect R-loop formation, chromatin accessibility and PolII loading?
2. Figures 2 and 3, ATRX KO only affect 53 DEGs, yet it is unclear how it downregulated 56 pathways. The DEGs should be listed, and gene sets correlate with 56 downregulated pathways need to be detailed analyzed and presented.
3. ATRX deficiency affects R-loops, genome stability and gene transcription in addition to alpha globin genes. Does CX-5461 stabilize R-loop in VNTR that is depended on VNTR variable (Figure 4), as ATRX targets tandem repeats and influences gene expression in a size-dependent manner?
4. ATRX KO mediated R-loop stabilization is not HBM VNTR specific, and its effect could be indirect and correlative. Do direct targeting and resolving VNTR R-loops (such as CRISPR-dCas9 targeting ATRX or RnaseH1 to VNTR region) rescue alpha globin expression and alleviate alpha thalassemia in ATR-X syndrome associated alpha thalassemia patient samples. Such experiments may be out of scope of current manuscript. However, authors should at least discuss its potential translation applications in the discussion section.

Version 1:

Reviewer comments:

Reviewer #1

(Remarks to the Author)

In the absence of being able to directly assess DNA damage at the VNTR, the authors have adequately addressed my other concerns. Perhaps there is a better title than the one currently provided, as the authors cannot conclusively show that there is genomic instability at the G-rich repeats. Beyond that, I have no further comments, and the authors should be congratulated on their well-performed study and nuanced interpretation of their data.

Reviewer #2

(Remarks to the Author)

All concerns have been addressed.

Reviewer #3

(Remarks to the Author)

1) The authors should clearly discuss the constraints of their model system. Contrary to statements from the authors, the study lacks in vivo data. All mechanistic findings rely on in vitro models (primary human CD34+ cells and HUDEP-2 cell lines) and do not constitute true in vivo analyses i.e., experiments performed in a living organism and cannot assess physiological and disease states. The authors should clarify in the abstract and discussion that findings were made in human cell culture models to eliminate misinterpretation about the nature of the experimental evidence.

2) Only gene-edited lines from healthy donors were used. Use of cells from ATR-X patients would bolster disease relevance.

3) The manuscript repeatedly overstates the novelty and clinical relevance of their findings. i.e. "uncovers a new molecular mechanism of human genetic disease" and "reveals an entirely new pathogenic mechanism underlying human genetic disease" and "disruption of this pathway to maintain genome integrity may underlie not only the developmental defects observed in ATR-X syndrome but also contribute to the role of ATRX mutations in ageing and cancer". The conclusion that this is a broadly applicable "entirely new" pathogenic mechanism is inappropriate given the focus on a single locus and in vitro cell systems. References to the finding as "an integrated model that unifies these mechanistic elements" may overstate the originality, since several cited studies have previously connected ATRX loss to G4 formation, R-loops, replication stress, and DNA damage signaling. The principal novelty here appears to be experimental demonstration at the alpha-globin VNTR with supporting cellular data but not a discovery of the general paradigm itself.

4) The authors state that their findings account for "several previously unexplained observations in ATR-X syndrome," including phenotypic heterogeneity and the subset nature of the affected cells. While the study offers support, the data are mainly correlative, and broader mechanistic explanations involving differentiation heterogeneity or additional modifying factors are not excluded. Overall, the study does not fully clarify what drives heterogeneity, nor does it provide definitive single-cell or clonal evidence for the underlying stochastic events triggering DNA damage in only a subset of ATRX-null cells. Requested revisions are discussed as avenues for future research. While the discussion attempts to bridge the gaps, much of this interpretation remains speculative and is not directly substantiated by the presented data.

5) The title's use of the plural "G-rich repeats" and generalization to "gene expression" may suggest a more global effect than was directly shown. Most functional and mechanistic evidence is at this single, disease-relevant VNTR, and while genome-wide correlations are discussed, only limited data are provided from other loci.

Reviewer #4

(Remarks to the Author)

The Authors have answered all concerns that I raised. The current revision including additional experiments performed significantly improves the manuscript and supports the key findings that ATRX regulates a-globin gene by preventing persistent R-loop formation. I have no further questions.

Reviewer #1 (Remarks to the Author):

This manuscript from Shen et al studies the role of ATRX in regulating transcriptional output from the α -globin locus. The main conclusion of this work is that ATRX enhances expression of the α -like globin gene HBM by resolving R-loop formation and DNA damage at a G-rich variable number tandem repeat (VNTR) DNA sequence near the gene promoter. Overall, the concepts presented are not novel per se, rather the strength of the study is that it confirms previous literature and presents a unified theory for ATRX function in regulating gene expression from the α -globin locus. The study itself is well done, using genetic and chemical perturbation to probe the role of the VNTR in HBM expression. However, the conceptual advances are modest, the scope in studying a single locus is limited, some claims are poorly supported, and some observations remain unexplained.

We thank the reviewer for their comments and for appreciating our work on the α -globin locus. The referee is correct in saying that the **concept** of how ATRX may cause human disease is not novel and indeed we and others have previously proposed this model. The difference is that here we have tested this hypothesis and proven the role of the putative G-rich repeats by deleting them in the presence or absence of ATRX.

By focussing in depth on the α -globin locus, we have established a mechanistic understanding of how mutations in *ATRX* may affect gene regulation. To extend this, we also examined the NME4 locus, another *ATRX* target, and observed effects consistent with those at the α -globin locus. In the revised manuscript, we have included a full list of the dysregulated genes affected by *ATRX* KO and found 45 out of 71 (63.4%) of them contain a GC-rich tandem or low-complexity repeats, suggesting this might be a general regulation rule of *ATRX*. Although the erythroid models used in this study limit the number of loci we could test, the mechanistic link we identify between *ATRX*, R-loop resolution, DNA damage, and transcriptional regulation is likely to extend to additional *ATRX* targets (as mentioned in lines 228-234), which could be explored in other models. As in numerous past studies, understanding mechanisms using this model locus have always extended to both normal and disturbed expression of many other mammalian genes (Higgs and Gibbons. The molecular basis of alpha-thalassemia: a model for understanding human molecular genetics. Hematol. Oncol. Clin. North Am. 2010).

Our work also goes beyond prior molecular descriptions by directly connecting these mechanisms to human disease, providing, to our knowledge, the first demonstration of this pathway's clinical relevance. We have now made these points more explicit in the revised manuscript (lines 64-65, 71-76, 276-281, 407-419).

The work follows up on a seminal finding from the same lab that *ATRX* plays a role in regulating genes near tandem repeats, with longer VNTRs resulting in a higher degree of gene downregulation upon *ATRX* loss (PMID 21029860). In the current study, the authors use CRISPR gene editing tools to delete the VNTR proximal to HBM and demonstrate that HBM expression is elevated and no longer dependent upon *ATRX*. This part of the study extends upon a study drawing similar conclusions for the HBM promoter in the context of a luciferase reporter assay (PMID 26991472).

The current study goes on to link ATRX to G-quadruplex binding, R-loop formation, DNA damage, and transcription. Many studies in the literature have demonstrated such connections (e.g., PMID 21029860, 25452430, 28487353, 30808951, 34162889, 35013239).

We agree that our work builds upon findings from the previous work and is consistent with other reports. Previous studies only examined or indicated an isolated aspect of ATRX function – such as G-quadruplexes (PMID 26991472, 25452430, 30808951, 34162889) or R-loops (PMID 28487353, 35013239), some in artificial *in vitro* systems, and have not connected these elements into a unified pathway to demonstrate how these biological processes interact in a physiological context. As discussed above, our study is distinguished by integrating these mechanisms and providing direct experimental test of the proposed mechanism *in vivo*. To highlight this, we have revised the discussion (lines 273-281) to clarify the novelty and significance of our work.

One conclusion from this work is that ATRX deficiency downregulates α -globin in a subset of cells exhibiting DNA damage. This part of the work is poorly supported, and the authors do not definitively demonstrate that the cells exhibiting signs of DNA damage are the same cells downregulating α -globin transcription. Further, the authors do not explain why loss of ATRX leads to stochastic changes in α -globin gene expression on a single cell level. These claims and observations require additional experimental support and analysis.

Thank you for this valuable comment. In response, we have performed additional single-cell analyses to ask whether α -globin downregulation and the increased DNA damage markers co-occur within the same individual cells. These new results are now presented in Fig. 2i and Extended Data Fig.3g, with further details provided in the responses below.

We acknowledge that the stochastic nature of gene expression changes in the absence of ATRX is intriguing. We propose that ATRX deficiency exerts cumulative effects on a subset of cells, as supported by our BFU-E colony assays in which only a fraction of colonies displayed gene downregulation, indicating that impairments in this subpopulation accumulate progressively during erythroid differentiation. Specifically, we suspect only cells that accumulate unresolved G4s/R-loops and replication stress are to be affected. The likelihood of a replication or transcription fork encountering these structures during a dynamic process introduces a stochastic element to the effect. Notably, ATRX is not the sole factor capable of destabilising G4s and R-loops; other helicases, such as FANCD1 DNA helicase, RecQ helicases, and DDX5, can also resolve these structures to release the cells from stress. Therefore, the affected subpopulation likely represents cells that lack ATRX to resolve G4s and retain G4s that cannot be resolved by other complementary helicases in time. With successive rounds of cell division and differentiation, consequences of these unusual structures are more likely to accumulate and impair DNA and chromatin, ultimately suppressing gene expression. We have now added this explanation in our revised Discussion section (line 371-386).

Specific Comments:

1. The authors state that only a subset of ATRX KO BFU-E clones show decreased HBM and HBA

expression. First, the authors are missing callout to Extended Data Fig. 1j, which shows this relationship better than the box plots (where information about single cell behavior is lost) chosen for the main figure Fig. 1f. This point should be better represented in the main figure, to show the expression level of HBM and HBA in individual BFU-E, compared to their ATRX expression level.

Thank you for highlighting this. We apologise for the previous omission, and we have now moved the panel previously shown as Extended Data Fig. 1j to the main figure (now Fig. 1f). This revised figure more clearly illustrates the expression levels of HBM and HBA in individual BFU-E clones relative to ATRX expression, as suggested.

2. Do all 53 downregulated genes in ATRX KO cells have VNTRs near their promoters or regulatory elements? Such genome-wide analysis would help to expand the scope of the study and the impact of the conclusions.

Thank you for this excellent suggestion. In response, we examined the occurrence of GC-rich tandem or low-complexity repeats (using the RepeatMasker track from UCSC) within 2 kb of all dysregulated genes identified in ATRX KO cells. Of the 71 dysregulated genes, we found that 45 have GC-rich repeats proximal to their promoters or regulatory elements. We also checked ATRX ChIP-seq data at these loci and found that ATRX binds all these regions. We have now included a full table summarising these in our revised manuscript (new Table 1) and have updated the text accordingly (see lines 228-235).

3. Lines 167-169 – links to DNA damage and HBM downregulation from the single cell data are weak. The argument is that (1) some ATRX KO cells have increased RNF168 expression; (2) some ATRX KO cells have increased gamma-H2AX staining; and (3) some ATRX KO cells with low HBM expression show an ontology term for histone H2A ubiquitination pathways. The authors have not confidently demonstrated that any of these observations are linked. Related, lines 313-315 claim in discussion that the work demonstrates increased H2A ubiquitylation in ATRX KO is overstated. The authors identify this pathway as an ontology term but they do not directly measure H2Aub.

Thank you for raising this important point. We agree that the connections between DNA damage and HBM downregulation were not fully delineated.

To address this, we have:

1. Assessed the expression of DNA damage response gene in the affected HBM^{low} subpopulation (new data in Fig. 2i and Extended Data Fig. 3g)
2. Directly measured H2Aub expression (new data in Fig. 2h)

As RNF168 expression is lowly expressed in our dataset, as shown in Fig. 2c, it was difficult to assess its contribution in the affected cells. However, while revisiting this point, we identified another DNA damage response gene, UIMC1 (also known as RAP80), which was more abundant and significantly upregulated in ATRX KO cells as shown below. RAP80 is a ubiquitin-binding protein that recognises ubiquitinated histones at DNA damage sites and facilitates BRCA1 recruitment to

the damaged sites, thereby mediating damage signalling and repair (Sobhian et al, Science 2007). Furthermore, a recent study identified RAP80 as a modulator to suppresses genomic abnormalities around the damaged site in transcription active regions (Yasuhara et al, Cell Reports 2022).

To examine its expression in the affected cells by colouring the cells by the expression level of UIMC1, we found that the affected HBM-low ATRX KO cells express high levels of UIMC1 (a, new data in Extended Data Fig. 3g), and this has been further supported by violin plot shown in (b, new data in Fig. 2i). This suggests that the affected subpopulation exhibits upregulated DNA damage signalling and enhanced DNA damage response pathway.

To check whether ATRX KO cells have global increase in H2A ubiquitination, we performed H2Aub immunofluorescent staining and found that a subset of ATRX KO cells showed elevated H2Aub levels.

These new results have been added to the revised manuscript (Fig. 2h), with text updated from line 168-177. We acknowledge that further work is needed to demonstrate locus-specific H2A ubiquitination in the affected subpopulation.

4. The authors do not directly measure DNA damage at the α -globin VNTR in the absence of ATRX but rather make inferences that DNA damage of this locus plays a role in its transcriptional repression. The location of DNA damage in the absence of ATRX should be assessed.

Thank you for raising this point. We agree that detecting the location of DNA damage around the VNTR at the α -globin locus would significantly strengthen the study. We have tried to address this and collaborated with experts in the DNA damage field. However, measuring locus-specific DNA damage is technically challenging, particularly in our model where damage occurs in only a small subset of cells. To pursue this, we collaborated with Dr. André Nussenzweig to apply END-seq, a method developed in his group to map DNA double-strand breaks (Canela et al, Molecular Cell 2016), as well as S1-END-seq to detect single-stranded DNA structures (Matos-Rodrigues et al, Molecular Cell 2022). Unfortunately, given the subtle nature of the effect in our model, we were unable to detect convincing signals at the α -globin locus, as shown below.

We also performed gammaH2AX ChIPmentation in our HUDEP models, which suggested a possible enrichment around the VNTR region (highlighted in blue below). However, we do not consider these results sufficiently robust and convincing to draw conclusion, likely due to the low frequency of affected cells in the population.

Therefore, to get around with this point, we instead performed a locus-specific CRISPR-guided DNA damage assay to link DNA damage signalling directly to local gene downregulation. Using this approach, we observed distance-dependent perturbation of gene expression around the damage site at the α -globin locus (Fig. 4f, unchanged, shown below for reviewer's reference).

Finally, in recognition of the importance of direct detection of locus-specific DNA damage, we have explicitly discussed this limitation in the revised Discussion (lines 386-390).

5. Out of curiosity, what happened to tubulin at the 4 hr time point in Extended Data Fig. 4c?

We apologise for the oversight, which resulted from a formatting issue during PDF export from PowerPoint. This has now been corrected, and we have carefully reviewed all figures to ensure consistency and accuracy in the revised manuscript.

Reviewer #2 (Remarks to the Author):

In this manuscript Shen et al examine ATRX function in gene regulation in an erythroid differentiation model. ATRX mutations are known to result in a mild form of alpha thalassemia. The goal of this paper is to develop a mechanistic understanding of ATRX function in this process. First the authors show that loss of ATRX results in reduced HBM expression only in a subset of cells. They further perform single cell RNA seq to show deregulation in a subset of cells and that a subset of these cells also exhibit gamma H2AX foci indicative of DNA damage. In addition, RNA sequencing analysis showed an upregulation of DNA damage response pathways. They develop a degron model where they can acutely deplete ATRX in HUDEP2 cells. Using these cells they show that ATRX binds to the G-rich VNTR region at the alpha globin locus in a transcription dependent manner and that loss of ATRX results in reduced HBM expression. Effects of ATRX loss are attenuated by deletion of the VNTR region. Finally they show that the VNTR region accumulates R-loops in the absence of ATRX. Interestingly introduction of DNA damage at these sites also recapitulates ATRX loss with respect to gene expression.

Overall this is a straightforward paper that is easy to follow and logical in narrative. However most of the conclusions of this paper have already been published previously by the same group or by several different groups in other journals as mentioned below. While the degron system itself is

novel, there is very little novelty in the experiments performed using this system. Overall this manuscript does not contribute significantly to advancing our understanding of how ATRX regulates gene expression beyond what we know already.

Specific comments:

1. Several groups have already shown that ATRX loss results in accumulation of DNA damage in different cell types. These include Wong et al in genome research 2010, Lovejoy et al plos genetics 2012, Huh et al JCI 2012. This is not new or unexpected.

It is true that prior studies (including Wong et al, Genome Research 2010, Lovejoy et al, PLoS Genetics 2012, and Huh et al, JCI 2012) have demonstrated that ATRX loss leads to DNA damage. Our study does not claim novelty in this respect; rather, it seeks to advance the field by experimentally testing the current speculation that ATRX-mediated DNA damage and G4/R-loop is the cause of disordered gene expression using the alpha-globin cluster as a model. Our work stands out by integrating these interconnected processes and demonstrating how ATRX coordinates non-canonical genomic structures, genome stability, and transcriptional output. Furthermore, our findings provide, to our knowledge, the first direct evidence connecting these events to pathogenic phenotype in a human genetic disorder.

To reflect this, we have revised the introduction (lines 64-65, 71-76) and discussion (lines 276-281, 407-419) to clarify the novelty and significance of our work.

2. Gene ontology analysis in a small set of genes is not very informative or accurate. Presence of one or two gene in a category can point to the enrichment of a pathway. The same conclusions can be obtained by just looking at the gene names.

Thank you for this valuable comment. In response, we have now included the full list of dysregulated genes by ATRX KO in Table 1 of the revised manuscript, along with information on their association with GC-rich repeats and ATRX binding. Notably, a closer inspection of the individual gene identities revealed additional evidence supporting upregulation of DNA damage response pathways. Notably, while revisiting this analysis and addressing parallel reviewer comment, we identified significant upregulation of UIMC1 (RAP80) in ATRX KO cells, as shown below. RAP80 is a well-established DNA damage response factor that recognises ubiquitinated histones at DNA damage sites and recruits BRCA1 for repair (Sobhian et al, Science 2007).

We also examined UIMC1 expression within the HBM-low ATRX KO cells and found a significant increase (shown below, new data in Fig. 2i), suggesting that these affected cells display robust activation of DNA damage signalling pathways.

3. Fig 4 - Does DNA damage actually occur at the VNTR upon ATRX loss and R-loop accumulation? Some genomic method to evaluate this is needed.

We agree that this is a very important point.

We have performed dot blot assay to confirm there is a genome-wide increase of R-loop accumulation in ATRX deficient cells (new data in Fig. 4b).

We agree that direct detection of DNA damage around the VNTR would significantly strengthen the study. We have attempted to address this by performing gammaH2AX ChIPmentation in our HUDEP models, which suggested a possible enrichment around the VNTR region (highlighted in blue below). However, we do not consider these results sufficiently robust and convincing to draw conclusion, likely due to the mild effect and low frequency of affected cells in the population.

We also collaborated with experts in the DNA damage field trying to address this. With Dr. André Nussenzweig, we applied END-seq, a method developed in his group to map DNA double-strand breaks (Canela et al, Molecular Cell 2016), as well as S1-END-seq to detect single-stranded DNA structures (Matos-Rodrigues et al, Molecular Cell 2022). Unfortunately, we were unable to detect convincing signals, likely due to the mildness and subset-specific nature of the DNA damage in our model.

Therefore, to get around with this point, we instead performed a locus-specific CRISPR-guided DNA damage assay to link DNA damage signalling directly to local gene downregulation. Using this approach, we observed distance-dependent perturbation of gene expression around the damage site at the α -globin locus (Fig. 4f, unchanged, shown below for reviewer's reference).

We have discussed this limitation in the revised manuscript (lines 386-390).

4. Fig 4c,d – Does inducing DNA damage cause R-loop formation at VNTRs given the recent reports of R-loops in DNA damage repair?

Thank you for bringing up the relationship between R-loops and DNA damage repair. In our current study, the experiments shown in the original Fig 4c,d were performed in the VNTR knockout model,

where the region is deleted and it was not possible to assess R-loop formation at the VNTR in response to induced DNA damage within this system.

We agree that this is an important mechanistic question, especially given recent studies reporting that R-loop formation near DNA double-strand breaks could facilitate DNA damage repair (e.g. Ngo et al, Nature Communications 2021; Girasol et al, PNAS 2023). Some of these findings have been described in non-physiological contexts following the use of DNA-damaging agents, and they found transcription at DSBs can promote local R-loop formation, which in turn provides binding platforms for DNA damage response factors.

Given the highly GC-rich (96%) nature of the VNTR at the α -globin locus and its active transcription under physiological conditions, we speculate that, in the absence of ATRX, R-loop formation at this site is primarily driven by transcriptional activity and the high GC content, rather than being the consequence of DNA damage. This interpretation is consistent with the previous data from our group (Nguyen et al, EMBO Rep 2017), which showed that R-loops formed at G-rich regions are transcription-dependent. We therefore propose that persistent G4 structures and R-loops in this context cause DNA damage, rather than being secondary to it. Nevertheless, investigating whether R-loop accumulation can also be further enhanced by DNA damage at the VNTR, potentially supporting a bidirectional relationship, is an important and interesting future direction. We have added a discussion of this point in the revised manuscript (lines 327-335) but believe that fully addressing this mechanistic question is beyond the current study's scope.

5. The title of the paper is misleading – it suggests that ATRX is able to resolve R-loops, when there is no evidence to suggest that ATRX functions in unwinding R-loop or G4 structures.

We appreciate the reviewer's point and acknowledge that we do not directly prove its unwinding or helicase activity. We have now revised our title to be 'ATRX loss couples genomic instability at G-rich repeats to dysregulation of gene expression'.

Reviewer #3 (Remarks to the Author):

This study by Shen et al investigates the molecular mechanism by which mutations in the chromatin remodeler ATRX cause α -thalassemia, a feature of the developmental disorder ATR-X syndrome. The authors propose a multi-step mechanism for the regulation of the α -globin gene: ATRX deficiency leads to the accumulation of R-loops and G-quadruplex (G4) structures at a specific G-rich variable number tandem repeat (VNTR) in the α -globin locus, which in turn induces local DNA damage and subsequent transcriptional silencing of the adjacent α -globin genes.

They employ complementary experimental systems including CRISPR-edited primary CD34+ hematopoietic stem cells and an inducible ATRX degon system in HUDEP-2 cells. The former approach allows for physiologically relevant analysis in primary cells and the latter enabled

temporal control of ATRX depletion. The scRNAseq analysis is particularly valuable, as it revealed the stochastic nature of the α -globin downregulation. The most compelling aspect of the study is the genetic rescue and recapitulation experiments.

Nevertheless, several weaknesses should be addressed:

1-Several figures lack statistical testing or only show representative data without quantification. The sample sizes for some experiments is low (some even at $n=2$) and appear limited for drawing any conclusion. The authors should include rigorous statistics to all quantitative assays: replicate numbers (replicates should be biological rather than technical), adjusted p-values, and clear indication of the statistical tests used.

We appreciate the emphasis on statistical robustness. All n values presented in the manuscript represent biological replicates. Each experiment was performed independently at least three times, except for the colony-forming unit assay using AAVS1 and ATRX KO CD34⁺ cells, where we analysed over 300 single colonies from two independent batches ($n = 2$ biological repeats). In this specific assay, gene expression was assessed at the level of individual colonies rather than bulk measurement between genotype groups, which we believe does not bias the conclusions drawn. We have carefully double-checked all statistical analyses in the study and included quantification for the representative data. Appropriate and rigorous statistical tests were applied across all datasets with $n \geq 3$, and detailed descriptions of these tests, as well as exact p-values, are now provided in the respective figure legends, as well as in the 'Statistical analysis' method section in the revised manuscript.

2- While the paper establishes that ATRX loss leads to R-loop accumulation and subsequent DNA damage, it remains unclear whether ATRX primarily resolves R-loops, which then stabilize G4s, or acts on both independently. Additional experiments, such as overexpressing RNase H1 to degrade R-loops, could clarify whether resolving R-loops alone prevents DNA damage and rescues α -globin expression.

Thank you for this insightful comment. To address it, we performed RNase H1 overexpression assay by transfecting RNase H1-GFP plasmid into ATRX degron cells and analysed gene expression in GFP⁺ cells ($n=3$). The results (new data in Fig. 4d) show a partial rescue of HBM expression upon R-loop degradation. This suggests ATRX modulates R-loops and G4 structures at least partially independently. Given the high GC content (96%) of the VNTR, G4 formation likely occurs intrinsically. In ATRX-deficient cells, transcription enhances both R-loop and G4 generation, leading cumulatively to chromatin instability and DNA damage. This new data strengthens our mechanistic model and we have now discussed this point in the discussion (lines 325-335).

3- The authors suggest that the mechanism may be general, citing downregulation of NME4, another VNTR-containing gene. However, this is based on limited evidence. A genome-wide analysis of ATRX binding at G-rich repeats, R-loop accumulation, and transcriptional effects upon ATRX loss are required to support claims of broader relevance.

We appreciate this suggestion and have systematically re-analysed our scRNA-seq dysregulated gene list from ATRX KO CD34+ cells for presence of GC-rich repeats and ATRX binding based on our ChIP-seq data. Remarkably, 45 out of 71 dysregulated genes contained nearby GC-rich tandem or low-complexity repeats bound by ATRX. We now present these data in new Table 1 and have expanded the text to discuss the potential broader applicability of our model beyond the α -globin locus (lines 228-235).

4- The paper does not fully address why only a subset of ATRX-deficient cells exhibit the phenotype. Factors such as differences in transcription levels, replication timing, or pre-existing epigenetic states at the locus should be investigated to provide deeper insight into the phenotype's variability. The authors should investigate whether transcriptional load, replication timing or local chromatin marks or histone variants predict which ATRX-null cells become HBMlow.

We agree this is an important point that needs further investigation. Ideally, this could be addressed by isolating the affected subpopulation and profiling their underlying molecular signatures; however, this is not technically feasible at present due to the stochastic nature of the affected cells. Instead, we have leveraged our single-cell RNA-seq dataset to define affected (HBM^{low}) and unaffected (HBM^{high}) subpopulations in ATRX KO sample and compare their transcriptional profiles. This analysis, described in Extended Data Fig. 3e (unchanged, shown below for reviewer's reference), was performed through GSEA to identify pathways distinguishing these two populations. We have provided the full pathway list and corresponding gene sets in a new supplementary table (Extended Data Table. 1) in the revised manuscript.

We also performed cell-cycle analysis at the single-cell level based on gene expression signatures and found that the affected HBM^{low} ATRX KO cells were predominantly in the G1 phase (Extended Fig.3b). This is consistent with the upregulation of DNA damage response pathways and elevated levels of γ H2AX observed in ATRX KO cells compared to controls. It is known that unrepaired DNA damage from G2/M phases is resolved during G1 through 53BP1 nuclear bodies, which often localize at chromosomal fragile sites (Lukas et al, 2011). ATRX has been reported to colocalize with 53BP1 nuclear bodies in G1 to facilitate the repair of these fragile regions (Pladevall-Morera et al, 2019). In our single-cell data, TP53BP1 expression was modestly increased in ATRX KO cells, though not significantly, supporting this connection. Collectively, these data suggest that ATRX-deficient cells accumulate DNA damage at fragile sites that require G1-phase repair, resulting in an enrichment of cells in this phase.

We propose that ATRX deficiency exerts cumulative, stochastic effects on a subset of cells. This is supported by our BFU-E colony gene expression data, in which only a fraction of colonies displayed α -globin downregulation, suggesting that impairments accumulate progressively during erythroid differentiation. We hypothesize that only cells with unresolved G4s/R-loops and associated replication stress become affected. Because the likelihood of a replication fork encountering these structures varies stochastically, only a subset of cells may experience persistent damage. Additionally, ATRX is not the sole helicase that resolves G4s and R-loops, other helicases such as FANCI, RecQ, and DDX5 may act redundantly. Thus, affected cells likely represent those in which ATRX loss coincides with insufficient compensation by other helicases, leading to cumulative DNA and chromatin instability that ultimately suppresses gene expression. We have now discussed this in the revised manuscript (lines 371-386).

5- While the paper cites studies supporting its model of R-loop and G4-mediated DNA damage, it does not adequately address alternative or conflicting mechanisms of ATRX-mediated α -globin

regulation. For instance, the role of ATRX in histone variant exchange, particularly the negative regulation of macroH2A incorporation at the α -globin locus, is well-documented in the literature. This study shows that ATRX loss leads to macroH2A accumulation at the HBA cluster, contributing to transcriptional repression. Although the current paper briefly mentions histone H2A ubiquitination pathways and cites Ratnakumar et al., 2012 in the discussion, it does not fully integrate or discuss this alternative pathway as a potential complementary or competing mechanism. A more explicit acknowledgment or comparison would improve the manuscript.

We appreciate this point regarding alternative or complementary mechanisms of ATRX-mediated α -globin regulation. Indeed, ATRX's role in macroH2A regulation at the α -globin locus, as described by Ratnakumar et al, 2012, represents an important pathway. We attempted to assess this mechanism in our model through macroH2A ChIPmentation, but the signal was not conclusive, likely due to the subtle transcriptional effects observed at the bulk population level. In our study, we identified upregulated DNA damage and H2A ubiquitination in ATRX KO cells, which is associated with transcriptional repression. These may be complementary to macroH2A-mediated regulation, as macroH2A has been implicated in DNA damage responses by accumulating at damaged regions and promoting local chromatin condensation, as summarised by Oberdoerffer and Miller, *Seminars in Cell & Developmental Biology*, 2023.

In response, we have expanded the Discussion (lines 358-369) to explicitly integrate our findings with the existing literature, emphasising that DNA damage-associated chromatin remodelling and macroH2A deposition likely represent parallel and potentially synergistic mechanisms of α -globin repression upon ATRX loss.

6- The temporal correlation between HBZP1 transcription and the appearance of an ATRX ChIP-seq peak at the $\psi\zeta$ VNTR, on its own, is not sufficient to conclude that ATRX binding is transcription-dependent. Differentiation can potentially change many variables—chromatin accessibility, histone modifications or variants, replication timing, DNA topology and the abundance of ATRX-binding partners. Any one of these could, in principle, be the true driver of ATRX recruitment, with HBZP1 activation occurring in parallel.

We agree that multiple factors may influence ATRX binding during differentiation. Although we have not comprehensively dissected these factors in this study, our previous work demonstrated that transcription alone at tandem repeats can drive ATRX binding (Nguyen et al, *EMBO Reports* 2017).

To avoid overinterpretation, we have revised our wording from 'transcription-dependent' to 'co-transcriptionally' (line 29) and 'correlated with transcriptional activation' (line 199) in the revised manuscript. We also refer to the study supporting transcription as a major driver of ATRX recruitment, while acknowledging the broader regulatory context in which these interactions occur.

7- Several findings consolidate or refine previously reported associations and consequently the study provides important, but at times incremental information (ATRX–VNTR binding, G4 propensity, macroH2A antagonism). The binding of ATRX at this VNTR was previously reported. The G4 aspect

has been described. The use of CX-5461 confirms, rather than discovers, the G4 component. MacroH2A and other epigenetic players are mentioned but not mechanistically explored; Ratnakumar et al. already implicated macroH2A in α -globin repression after ATRX loss.

Thank you for recognising the logical progression and importance of our study. While individual elements such as ATRX-VNTR binding, G4 structure formation, and macroH2A regulation have been reported, they were studied separately in different contexts. Our work uniquely integrates and expands these mechanisms into a unified pathway that explains how ATRX loss leads to α -thalassemia in a physiological context. In addition, as pointed out above, previous reports have hypothesised about how the VNTR might underlie this unusual form of α -thalassaemia. Here we have experimentally tested this and shown that in the absence of the VNTR, the alpha genes are no longer downregulated when ATRX is removed. In addition, if we artificially induce DNA damage at the position of the VNTR this does downregulate alpha globin expression. Specifically, our study provides mechanistic insights into several previously unresolved questions in the field:

1. Why ATRX germline mutation leads to only a subtle α -thalassemia phenotype in ATR-X syndrome patient,
2. Why the gene downregulation occurs in a distance-dependent manner, and
3. Why ATRX mouse models fail to fully recapitulate human α -thalassemia.

These aspects could not be explained by earlier studies in isolation. Our contribution thus represents the first integrated *in vivo* demonstration linking ATRX, VNTRs, R-loops, G4s, DNA damage and chromatin dynamics within the context of human disease. We have emphasized this conceptual advance and its implications in the revised introduction (lines 64-65, 71-76) and discussion (lines 276-281, 407-419).

Reviewer #4 (Remarks to the Author):

In this manuscript, Shen et al. demonstrated that ATRX is an essential regulator for alpha-globin gene expression through resolving R-loop formation and preventing DNA damage at the G-rich VNTR located upstream of HBM gene. The pioneer work from Dr. Higg's lab revealed that in ATR-X syndrome, mutation of ATRX resulted in α -thalassemia through downregulated alpha-globin genes. In this study, they provide a unique mechanism by which ATRX specifically regulates alpha-globin gene expression via preventing persistent R-loop formation in upstream of HBM gene. The manuscript is well written, and the conclusions are generally supported by the experiments conducted by the authors. However, there are some comments that need to be addressed to strengthen and improve the manuscript.

1. Figure 1 showed that ATRX KO efficiency is more than 70%, yet HBM is reduced only in very small fraction of cells. Majority of ATRX KO cells do not exhibit reduction of alpha-globin genes. Are variable length and repeat # of VNTR involved in regulation of ATRX dependent alpha globin regulation? Does VNTR variable affect R-loop formation, chromatin accessibility and PolIII loading?

Thank you for this insightful question. We agree that the length of the VNTR likely plays a key role in ATRX-dependent regulation of α -globin expression. As reported previously by our group (Law et al, Cell 2010), the degree of α -globin downregulation in ATR-X syndrome patients correlated with VNTR size. In the current study, we provide a mechanistic model to explain this observation - namely, that longer VNTRs are more prone to forming R-loops and G4 structures at the locus, which in turn can disrupt local chromatin organisation and transcription of nearby genes.

We have determined the VNTR length in all donor samples used in this study. However, due to variability in ATRX knockout efficiency across donors and the relatively modest extent of α -globin downregulation, the current dataset does not allow a robust correlation between VNTR length and gene repression. We fully agree that establishing isogenic models carrying VNTRs of different lengths would be valuable to directly test whether VNTR size affects chromatin accessibility and Pol II loading. Unfortunately, due to the VNTR's extremely high GC content (~96%), precise CRISPR editing across this region remains technically challenging. For this reason, we instead generated a model in which the entire VNTR region was deleted to assess its functional contribution.

In support of the importance of repeat length, an earlier study from our group (Nguyen et al, EMBO Reports 2017) demonstrated that ATRX recruitment to G-rich tandem repeats depends on repeat number, orientation, and transcriptional activity. The transcribed repeats form R-loops whose abundance correlates with ATRX binding. In the present work, we show that similar R-loop-dependent mechanisms occur at the α -globin VNTR, leading to local DNA damage and transcriptional repression.

The observation that α -globin downregulation occurs only in a subset of ATRX KO cells likely reflects stochastic effects of ATRX loss at the single-cell level. At present, it is technically infeasible to assess VNTR length heterogeneity within individual cells to determine whether the affected subset carries longer repeats during differentiation. We have tried to determine the VNTR length in the affected BFU-E colonies, however the DNA generated via whole genome amplification from a single colony does not yield good enough quality for such detection. We recognise this as an important direction for future investigation and have expanded our discussion to highlight this point (lines 291-296, 390-394).

2. Figures 2 and 3, ATRX KO only affect 53 DEGs, yet it is unclear how it downregulated 56 pathways. The DEGs should be listed, and gene sets correlate with 56 downregulated pathways need to be detailed analyzed and presented.

We appreciate the suggestion to clarify the connection between DEGs and pathway analysis. Accordingly, we have included the full list of 53 DEGs in the new Table 1, along with details on their association with GC-rich repeats and ATRX binding. The 56 dysregulated pathways were identified through GSEA analysis comparing HBM^{high} and HBM^{low} populations, where the number of DEGs is substantially larger. This analysis aimed to capture the pathways differentially regulated between the unaffected and affected subpopulations, with strategies outlined in Extended Data Fig. 3e (unchanged, shown below for reviewer's reference).

We have now included the full pathway list and corresponding gene sets in a new supplementary table (Extended Data Table. 1) in the revised manuscript.

3. ATRX deficiency affects R-loops, genome stability and gene transcription in addition to alpha globin genes. Does CX-5461 stabilize R-loop in VNTR that is depended on VNTR variable (Figure 4), as ATRX targets tandem repeats and influences gene expression in a size-dependent manner?

Thank you for highlighting this important point. It is correct that ATRX's effects on R-loops are potentially influenced by VNTR sequence properties. However, our current models do not include designed variants of the VNTR length and due to technical and material limitations, we were unable to perform DRIP analysis on CX-5461 treated samples, as these experiments require substantially larger cell numbers.

We fully agree that assessing the impact of VNTR length on R-loop formation and chromatin dynamics in appropriate models would provide valuable mechanistic insight. This point has now been discussed in the revised discussion (lines 291-296).

4. ATRX KO mediated R-loop stabilization is not HBM VNTR specific, and its effect could be indirect and correlative. Do direct targeting and resolving VNTR R-loops (such as CRISPR-dCas9 targeting ATRX or RnaseH1 to VNTR region) rescue alpha globin expression and alleviate alpha thalassemia in ATR-X syndrome associated alpha thalassemia patient samples. Such experiments may be out of scope of current manuscript. However, authors should at least discuss its potential translation applications in the discussion section.

Thank you for this thoughtful suggestion. We agree that direct resolving R-loops would provide highly informative rescue experiments. To this end, we performed RNase H1 overexpression experiments

in our inducible ATRX degron model and observed a partial rescue of HBM expression (new data in Fig. 4d). This result supports the idea that R-loop accumulation contributes functionally to α -globin repression upon ATRX loss. As ATR-X syndrome cases are very rare, we have little access to patient-derived samples.

Sincerely,

Douglas Higgs
Emeritus Professor, FRS, FMedSci
E: doug.higgs@imm.ox.ac.uk

Richard Gibbons
Professor, FMedSci
E: richard.gibbons@ndcls.ox.ac.uk

Reviewers' comments:

Reviewer #1 (Remarks to the Author):

In the absence of being able to directly assess DNA damage at the VNTR, the authors have adequately addressed my other concerns. Perhaps there is a better title than the one currently provided, as the authors cannot conclusively show that there is genomic instability at the G-rich repeats. Beyond that, I have no further comments, and the authors should be congratulated on their well-performed study and nuanced interpretation of their data.

We thank the reviewer for their positive feedback. In response to the suggestion about the title, we have revised it to more accurately reflect the scope of our work: "*ATRX loss couples genome instability at a G-rich repeat to dysregulation of human alpha-globin expression*".

Reviewer #2 (Remarks to the Author):

All concerns have been addressed.

Reviewer #4 (Remarks to the Author):

The Authors have answered all concerns that I raised. The current revision including additional experiments performed significantly improves the manuscript and supports the key findings that ATRX regulates a-globin gene by preventing persistent R-loop formation. I have not further questions.

Reviewer #3 (Remarks to the Author):

1) The authors should clearly discuss the constraints of their model system. Contrary to statements from the authors, the study lacks *in vivo* data. All mechanistic findings rely on *in vitro* models (primary human CD34+ cells and HUDEP-2 cell lines) and do not constitute true *in vivo* analyses i.e., experiments performed in a living organism, and cannot assess physiological and disease states. The authors should clarify in the abstract and discussion that findings were made in human cell culture models to eliminate misinterpretation about the nature of the experimental evidence.

We appreciate the importance of clarifying the system used. In our initial terminology, "*in vivo*" referred to live cells in their native chromatin context (distinct from reconstituted systems), which is conventionally used in a specific field of chromatin biology. However, to prevent any confusion for a broader readership, we have removed the term "*in vivo*" throughout the manuscript and clarify that our findings were obtained from human cell culture models.

2) Only gene-edited lines from healthy donors were used. Use of cells from ATR-X patients would bolster disease relevance.

We appreciate the reviewer's suggestion to use living organisms such as ATR-X patient samples. However, there is currently no suitable *in vivo* system that could provide additional mechanistic insight beyond what we have shown in this study.

As noted in the manuscript, the mouse α -globin cluster lacks the G-rich VNTR that underlies the ATRX-associated phenotype in humans. Indeed, mutations of *ATRX* in the mouse, do not cause alpha thalassaemia (Tillotson *et al. Hum Mol Genet* 2023) and we have the haematological data from a mouse model of ATRX (Tillotson *et al.* unpublished) showing normal readout of red blood cell parameters which we could include in the current manuscript if required. Therefore, mouse models cannot reproduce this aspect of the disease.

Regarding ATR-X patient samples, ATR-X syndrome is exceptionally rare – with only ~200 reported cases worldwide. Acquisition of adequate patient material is therefore unfortunately impractical. Of note, we have previously analysed two affected brothers and published the cellular heterogeneity in their erythroid cells (Truch J *et al. Nat Commun* 2022). Given the scarcity of patient samples, in the current study we established a physiologically relevant model using primary CD34+ cells from healthy donors, which faithfully recapitulates the key molecular and phenotypic features of the disease.

In addition, to understand the underlying molecular mechanisms to explain the observations in patients, we established and employed a HUDEP-2 degron cell model to perform two key experiments that directly establish pathogenic relevance: 1. Deletion of the VNTR abolishes the ATRX-loss phenotype. 2. Targeted induction of DNA double strand break at this site, remote from the alpha genes, reinstates the phenotype in a pattern identical to that seen when ATRX is mutated. These demonstrate experimentally for the first time that the G-rich VNTR itself is causally linked to the disease mechanism. Such gene-editing experiments are not feasible or ethically permissible in humans *in vivo*.

3) The manuscript repeatedly overstates the novelty and clinical relevance of their findings. i.e. "uncovers a new molecular mechanism of human genetic disease" and "reveals an entirely new pathogenic mechanism underlying human genetic disease" and "disruption of this pathway to maintain genome integrity may underlie not only the developmental defects observed in ATR-X syndrome but also contribute to the role of ATRX mutations in ageing and cancer". The conclusion that this is a broadly applicable "entirely new" pathogenic mechanism is inappropriate given the focus on a single locus and *in vitro* cell systems. References to the finding as "an integrated model that unifies these mechanistic elements" may overstate the originality, since several cited studies have previously connected ATRX loss to G4 formation, R-loops, replication stress, and DNA damage signalling. The principal novelty here appears to be experimental demonstration at the alpha-globin VNTR with supporting cellular data but not a discovery of the general paradigm itself.

We appreciate the reviewer's feedback on these points and the opportunity to clarify this. In the revised manuscript, we have revised the wording to make sure that the novelty and broad applicability are appropriately stated (lines 34, 303-305, 416-418).

While we acknowledge that previous studies have individually associated ATRX with certain aspects of G4, R-loops, or replication stress, none have, to our knowledge, experimentally

connected them and demonstrated an integrated causal mechanism linking these features to a disease phenotype. Our study provides a direct experimental validation of this model at the human α -globin locus. This extends prior hypotheses into experimentally proven causality. We have explicitly noted this significance in the Discussion (lines 270-280).

To our knowledge, this mechanism of dysregulation of gene expression mediated by a long-range VNTR in a human genetic disease context is novel. To make this specific point clearer, we have clarified it in the revised Abstract and Discussion (lines 34, 416-418). We would like to emphasise that a very high proportion of the principles underlying normal gene regulation and how this is perturbed in human genetic disease were first discovered at the globin gene loci and subsequently found to be generally true, including the original identification of *ATRX* gene itself (Gibbons, Picketts *et al. Cell* 1995).

4) The authors state that their findings account for "several previously unexplained observations in ATR-X syndrome," including phenotypic heterogeneity and the subset nature of the affected cells. While the study offers support, the data are mainly correlative, and broader mechanistic explanations involving differentiation heterogeneity or additional modifying factors are not excluded. Overall, the study does not fully clarify what drives heterogeneity, nor does it provide definitive single-cell or clonal evidence for the underlying stochastic events triggering DNA damage in only a subset of ATRX-null cells. Requested revisions are discussed as avenues for future research. While the discussion attempts to bridge the gaps, much of this interpretation remains speculative and is not directly substantiated by the presented data.

We thank the reviewer for these comments. We recognise that the working model is not yet fully delineated and that additional studies will be required for a complete mechanistic picture, as discussed as future avenues in the manuscript. Nevertheless, we believe the current data provide clear mechanistic insights into several previously unexplained observations, supported by direct experimental evidence:

1. Why do ATR-X syndrome patients present only a subtle α -thalassemia phenotype?

Our scRNA-seq analysis demonstrates that α -globin downregulation occurs only in a subset of ATRX-deficient cells, rather than as a uniformly mild reduction across all cells. This conclusion aligns very well with findings in patient-derived cells (Truch *et al, Nat Commun* 2022).

2. Why the gene downregulation occurs in a distance-dependent manner?

We show that gene dysregulation depends on the VNTR and on secondary R-loop formation and DNA damage, which together account for why genes located closer to the VNTR are more strongly affected.

Importantly, two key experiments, the VNTR deletion and targeted DNA damage induction at the α -globin cluster, directly test causality and substantiate the proposed model. These findings establish a direct link between ATRX loss, VNTR instability, and downstream transcriptional consequences.

Regarding the broader mechanistic explanation of phenotypic heterogeneity, we agree that multiple factors could contribute. However, differentiation heterogeneity appears unlikely to be a major driver: BFU-E colony assays show comparable morphology and colony numbers between ATRX-null and control cells (Extended Data Fig. 1h-i), indicating that the differentiation is not significantly perturbed. We have emphasised this point in the revised manuscript (line 375). Instead, our data suggest heterogeneity arises from molecular events at the G-rich VNTR. As discussed in the manuscript (lines 371-386), only cells that accumulate unresolved G4 /R-loops and replication stress exhibit transcriptional defects. The probability of replication or transcription forks encountering these structures introduces a stochastic element, potentially also modulated by other helicases in addition to ATRX. We agree this is an important direction for future research, however, the inherently stochastic nature of the process presents technical challenges for obtaining locus-specific evidence using current models. As noted as a limitation and future direction in the Discussion (lines 387-390), a more robust model with a stronger and more consistent phenotype will likely be required to address these questions definitively.

5) The title's use of the plural "G-rich repeats" and generalization to "gene expression" may suggest a more global effect than was directly shown. Most functional and mechanistic evidence is at this single, disease-relevant VNTR, while genome-wide correlations are discussed, only limited data are provided from other loci.

We appreciate this feedback and have revised the title to better reflect the study's focus. The new title, "*ATRX loss couples genome instability at a G-rich repeat to dysregulation of human alpha-globin expression*" more accurately captures the scope of the work.

We also thank the reviewer for recognising the 'functional and mechanistic evidence at this single, disease-relevant VNTR' at the α -globin locus. As mentioned above, this locus has long served as a foundational model for understanding gene regulation, and our study provides a mechanistic framework at this site that may guide future investigations into whether similar principles apply elsewhere in the genome.